⊘ | **Open Peer Review** | Antimicrobial Chemotherapy | Research Article

# Repurposing of the antimalarial agent tafenoquine to combat MRSA

Pengfei She,[1] Yifan Yang,[1] Linhui Li,[1] Yimin Li,[1] Shasha Liu,[1] Zehao Li,[1] Linying Zhou,[2] Yong Wu[2]

**ABSTRACT** *Staphylococcus aureus* is an important human pathogen that often leads to hospital-acquired and community-acquired infections. The appearance of methicillin-resistant *S. aureus* (MRSA) has made the treatment extremely challenging. There is an urgent need to develop new antimicrobial agents to combat MRSA infections. In this study, through high-throughput screening, we found that tafenoquine (TAF), an antimalarial agent, exhibited antimicrobial efficacy against MRSA and its highly resistant phenotypes of biofilm and persister cells. No resistant mutation emerged by consecutive sub-minimal inhibitory concentration of TAF induction. Mechanistic studies by fluorescent probes of SYTOX Green/DiSC3(5), molecular dynamic simulations, electron microscopy, and proteomic analysis revealed that TAF exerts antimicrobial activity mainly through selective bacterial cell membrane disruption. In addition, TAF exhibited an effective antimicrobial effect in a subcutaneous abscess model *in vivo* with a good toxicity profile. In conclusion, TAF may have the potential to be developed as a new antimicrobial agent to treat refractory MRSA infections.

**IMPORTANCE** This study represents the first investigation into the antimicrobial effect of TAF against *S. aureus* and its potential mechanisms. Our data highlighted the effects of TAF against MRSA planktonic cells, biofilms, and persister cells, which is conducive to broadening the application of TAF. Through mechanistic studies, we revealed that TAF targets bacterial cell membranes. In addition, the *in vivo* experiments in mice demonstrated the safety and antimicrobial efficacy of TAF, suggesting that TAF could be a potential antibacterial drug candidate for the treatment of infections caused by multiple drug-resistant *S. aureus*.

**KEYWORDS** methicillin-resistant *Staphylococcus aureus*, antimicrobial agents, drug repurposing, tafenoquine

S taphylococcus aureus (*S. aureus*) is a notorious Gram-positive coccus that usually colonizes the nasopharynx (1). *S. aureus* is a major pathogen that causes infections of the skin and soft tissues, lung, respiratory tract, or surgical sites, resulting in cardiovascular and intravenous indwelling device infections (2). In recent years, *S. aureus* bacteremia has become a serious threat because of its rising morbidity with 10%–30% mortality (2–4). Methicillin-resistant *S. aureus* (MRSA) was first identified in 1961, and in the following decade, outbreaks occurred in hospitals worldwide (5), with mortality from MRSA infection exceeding acquired immune deficiency syndrome, according to the Centers for Disease Control and Prevention (6). Subsequently, vancomycin (VAN)-intermediate and VAN-resistant *S. aureus* were reported in 1997 and 2002, respectively. Although they did not cause an epidemic, the treatment of resistant strains of *S. aureus* has become a serious challenge (7, 8).

The biofilm and persister cells are tolerant to antimicrobial agents and they often result in the failure of clinical therapy (9). Biofilms are extracellular complex structures

Address correspondence to Yong Wu, wuyong_zn@csu.edu.cn.

Pengfei She and Yifan Yang contributed equally to this article. Author order was determined by drawing straws.

The authors declare no conflict of interest.

See the funding table on p. 18.

composed of populations of microorganisms attached to the surface of the substrate, and bacteria within the biofilm are highly adherent and resistant, evading the host's immune response and avoiding antibiotic killing (10). Persister cells are small subsets of cells with non-heritable phenotypic variations. Persister cells are slow growing or dormant and can tolerate some high antibiotic concentrations and cause recurrence of infection (11). Therefore, it is necessary to explore the antimicrobial agent against the *S. aureus*-resistant phonotypes of biofilm and persister cells.

Antibacterial agents that selectively target the bacterial cell membrane are a promising strategy to overcome resistance. Because the bacterial cell membrane disruptors could act without entering into the inner cell. Moreover, the cell membrane disruptors can circumvent common bacterial resistance mechanisms such as enzymatic degradation and efflux pump (12, 13). Membrane-active antimicrobial agents are usually lipophilic, which allows better interaction with bacterial membranes, and disrupts the physiological function and physical integrity. It is undoubtedly that cell membranes are crucial for bacteria to maintain cellular homeostasis and perform metabolic activities. Although persister cells can escape conventional antibiotic killing by drastically downregulating the biosynthetic activities, they still need to convert energy in some way to maintain cell viability. Usually, this activity is dependent on the cell membrane (14). Therefore, the membrane-active antimicrobial agent is a promising way to combat persister cells.

In this period of bacterial resistance outbreaks and the high cost of new antibiotic development, drug repurposing is an attractive alternative to the discovery of antimicrobial agents. The pharmacological and toxicological information of the repurposed drug are well characterized; hence, the development costs could be reduced (15). For example, drug repurposing can save around 7 years and 15% of the costs compared to developing a new antibiotic by conventional way, and the known drug information greatly reduces the risk of failure (16).

The phenotype-based screening has emerged as an effective approach to antimicrobials discovery. Compared with mechanism-based screening strategies, it can be more effective in discovering compounds that target particular pathogens. Although the mechanism of these candidates is usually unknown, their known functions can serve as clues for mechanistic studies (17). Tafenoquine (TAF) is an 8-aminoquinoline that was first approved for the treatment of malaria in 2018 (18). The exact mechanism of TAF against plasmodium is unknown, but the process may involve interference of cytochrome C reductase leading to mitochondrial membrane depolarization, as well as the production of reactive oxygen species (ROS) (18). The bioactive effects of TAF against parasites, fungi, and *Mycobacterium tuberculosis* have been previously reported; however, its efficacy against other bacteria is unknown (19, 20). In the present study, using high-throughput screening, we identified TAF to be a potential antibacterial candidate against MRSA. Further, we investigated the *in vitro* and *in vivo* antibacterial activity as well as mechanisms of TAF in depth.

## MATERIALS AND METHODS

### Bacterial strains, culture conditions, and reagents

Type strains of *S. aureus* (ATCC 43300, USA300, ATCC 29213, and ATCC 25923), *Enterococcus faecalis* ATCC 29212, and *Enterococcus faecium* ATCC 19434 were purchased from the American Type Culture Collection (ATCC). *S. aureus* Newman was provided by Min Li (Shanghai Jiaotong University, Shanghai, China). *S. aureus* LZB1 and other clinical isolates were identified in the Third Xiangya Hospital of Central South University (Changsha, China) and identified by VETIK 2 Compact (bioMerieux, France) and MALDI-TOF (Bruker, Germany) (21). *Staphylococcus epidermidis* ATCC 12228 and RP62A were provided by Di Qu (Laboratory of Medical Molecular Virology, Shanghai Medical College, Fudan University, Shanghai, China). Other Gram-negative bacteria, including *Klebsiella pneumoniae* ATCC 700603 and *Escherichia coli* ATCC 25922, were provided by

Juncai Luo (Tiandiren Biotech, Changsha, China). *Pseudomonas aeruginosa* PAO1 was provided by Minqiang Qiao (College of Life Sciences of Nankai University, Tianjin, China). Gram-positive cocci of Staphylococcus and Enterococcus were cultured in tryptic soy broth (TSB) and brain-heart infusion (BHI) broth, respectively. Gram-negative strains were cultured in Luria-Bertani (LB) broth. All media were purchased from Solarbio (Shanghai, China). TAF and other antibiotics were purchased from MedChem Express (New Jersey, USA) and dissolved in deionized water or dimethyl sulfoxide (DMSO) according to the manufacturer's instructions.

## Antimicrobial susceptibility test

According to the recommendations of the Clinical & Laboratory Standards Institute (22), the minimal inhibitory concentration (MIC) of drugs was determined using the standard microdilution. First, the bacteria were grown to log phase and diluted with Mueller–Hinton (MH) II broth to $1.5 \times 10^6$ CFU/mL. Then, 50 µL of serially diluted reagents with 50 µL of the bacterial suspension were added to a 96-well plate. After incubating at 37°C for 16–18 h, the MIC was defined as the lowest concentration that inhibited visible bacterial growth. The minimum bactericidal concentration (MBC) was defined as the lowest concentration with visible colony growth on plates.

## High throughput screening

To find the antimicrobial molecules against *S. aureus*, the FDA-approved molecular library (Cat. No.: HY-L022, MedChem Express, New Jersey, USA) was selected with a storage concentration of 10 mM in DMSO. In the initial screening process, the log-phased *S. aureus* USA300 was adjusted to 0.5 McFarland (McF) with MH broth, then 99 µL of the bacterial suspension with 1 µL compound was mixed and added to a 96-well plate (~100 µM per well), and $OD_{630\ nm}$ was detected after 16–18 h incubation at 37°C. For the second screening procedure, the compound concentration was set as 50 µM and six hits were selected after removing the well-studied antimicrobials. After determining the MICs and MBCs of these candidates, TAF was ultimately selected for further study (23).

## Time-inhibiting and killing kinetics

Overnight culture of *S. aureus* was diluted to $1 \times 10^6$ CFU/mL with MH broth and incubated at 37°C 180 rpm in the presence of indicated concentrations of TAF. $OD_{630\ nm}$ was measured and the colony-forming unit (CFU) was counted at the particular time points of 0, 2, 4, 8, 12, and 24 h, respectively. DMSO (1%, vol/vol) was used as a control (24).

## Resistance-inducing assay

MICs of TAF, ciprofloxacin (CIP), and rifampin (RFP) against *S. aureus* were determined as described above on the first day. The bacterial suspension in sub-MIC ($1/2 \times$ MIC) was 1,000-fold diluted with MH broth for the next day's MIC determination. After 15 days of passages, the CIP- and RFP-induced resistant *S. aureus* strains of the last passage were used to detect the cross-resistance by TAF (24).

## Checkerboard dilution assay

Log phase grown *S. aureus* ATCC 43300 was diluted to $1 \times 10^6$ CFU/mL with MH broth. Fifty microliters of the bacterial suspension was mixed in equal volumes with serially diluted drugs into an "8 × 8" checkerboard. After incubation at 37°C for 16–18 h, the fractional inhibitory concentration index (FICI) between TAF and conventional antibiotics was calculated as follows:

$$FICI = \frac{MIC_{A(combination)}}{MIC_{A(alone)}} + \frac{MIC_{B(combination)}}{MIC_{B(alone)}}$$

FICI ≤0.5 indicates synergism; 0.5 < FICI < 1 indicates partial synergism; FICI = 1 indicates addition; 1 < FICI ≤ 4 indicates indifference; and FICI >4 indicates antagonism (25).

## Persister cells killing assay

*S. aureus* was grown for 24 h at 37°C and 180 rpm to the stationary phase to obtain persister cells (25). The persister cells were washed three times with 1 × phosphate-buffered saline (PBS) (pH 7.4), and adjusted to $OD_{630}$ = 0.2. Then, the persister cells were treated with 16 or 10 × MIC of VAN, daptomycin (DAP), gentamicin (GEN), and ciprofloxacin (CIP) to confirm the status of the persistant. For the persister cells killing assay, indicated concentrations of TAF were added into the persister cells, and the CFUs at the time point of 0, 2, 4, and 6 h were counted by fold dilution, respectively. DMSO (1%, vol/vol) was used as a control.

## Antibiofilm activity of TAF

*S. aureus* ATCC 43300 was incubated overnight in the stationary phase, and the bacterial suspension was diluted at 1:100 with TSB. For biofilm inhibition, the suspension was mixed with indicated concentrations of TAF and added to a 96-well plate. After incubation at 37°C for 24 h, the planktonic cells were carefully removed and the biofilm was washed with sterile saline. Then, the biofilm was fixed with methanol and stained with 0.25% (wt/vol) crystal violet (CV) for 15 min and washed three times with sterile saline. The dye attached to the biofilm was dissolved in 95% ethanol for 20 min, and the biomass of the biofilm was quantified by measuring $A_{570\ nm}$. For biofilm eradication, the bacterial suspension was allowed to form biofilm in a 96-well plate by incubation at 37°C for 24 h as described above. Then, the biofilm was washed and incubated with fresh TSB in the presence of indicated concentrations of TAF for another 24 h. CV staining was performed to quantify the biomass of biofilm. In addition, a fold dilution assay on agar plates was used to count the planktonic cells (26).

## Ultrastructure observation by electronic microscopy (EM)

Twenty milliliters of fresh TSB was mixed with 10 µL of bacterial suspension of *S. aureus* ATCC 43300, then cultured to the log phase. The cultures were centrifuged at 4,000× *g* for 8 min, and the sediment was washed twice with sterile saline. The bacterial cells were resuspended to McF = 0.5 with fresh TSB containing 5 × MIC of TAF or DMSO. After incubating at 37°C and 180 rpm for 1 h, the bacteria were centrifuged at 4°C and 4,000× g for 8 min, and the sediment was transferred to a 1.5 mL centrifuge tube and washed with 1 mL PBS and fixed with 2.5% glutaraldehyde. For scanning EM (SEM), the fixative was removed with PBS and dehydrated in increasing concentrations of ethanol (25, 50, 60, 70, 80, 90, and 100%) for 10 min for each solution. After drying, the samples were covered with gold palladium. For transmission EM (TEM), contrast was performed with 2% uranyl acetate and 2% osmium tetroxide in the dark. The material was then passed through an increasing series of ethanol and embedded in epoxy resin. Sections were prepared from the resin, placed on a copper grid, and stained for lead citrate. After sample preparation, cellular ultrastructure was observed using SEM and TEM (HITACHI, Tokyo, Japan), respectively (27).

## Bacterial membrane permeability assay

Log phase grown *S. aureus* ATCC 43300 was centrifuged at 4,000× g for 8 min, and the sediment was rinsed three times with 5 mM HEPES (pH 7.2) and resuspended to $OD_{630}$ = 0.2. After incubation in the dark with 2 µM of SYTOX Green for 15 min, the mixture was added to a black 96-well plate with indicated concentrations of TAF. The fluorescence intensity was consecutively measured every 5 min for a total of 30 min with excitation wavelength ($\lambda_{ex}$) and emission wavelength ($\lambda_{em}$) of 485 and 525 nm,

respectively. Melittin and 1% (vol/vol) DMSO were used as positive and negative controls, respectively (28).

## Membrane depolarization assay

*S. aureus* ATCC 43300 suspension was prepared as described above, except that HEPES buffer contained 5 mM glucose and 100 mM KCl. DiSC3(5) with a final concentration of 2 µM was incubated with the bacterial suspension for 45 min in the dark and then mixed with indicated concentrations of TAF. Fluorescence intensity was measured every 30 s for a total of 5 min with the $\lambda_{ex}$ and $\lambda_{em}$ of 622 and 670 nm, respectively. Melittin and 1% (vol/vol) DMSO were used as positive and negative controls, respectively (23).

## ROS detection

The overnight growth of *S. aureus* ATCC 43300 was washed and suspended to $OD_{630}$ = 0.5 with PBS. The suspension was incubated with H2DCFDA (10 µM) for 30 min in the dark. Then, 90 µL mixture and 10 µL TAF were added to black 96-well plates and incubated for 30 min. Fluorescence intensity was measured with the $\lambda_{ex}$ and $\lambda_{em}$ of 488 and 525 nm, respectively. Melittin and 1% (vol/vol) DMSO were used as positive and negative controls, respectively (23).

Fluorescence probes HKSOX-1, HKperox-2, and HKOH-1r (MedChem Express, New Jersey, USA) were used to evaluate the levels of specific reactive species of superoxide anion ($O_2 \bullet -$), hydrogen peroxide ($H_2O_2$), hydroxyl radicals ($\bullet OH$), respectively. As described previously, the bacterial suspension was co-incubated with 10 µM fluorescent probes for 30 min and then mixed with indicated concentrations of TAF, the fluorescence intensities were measured after another 30 min incubation, respectively, with λex/λem = 500/520 nm, 520/543 nm, and 500/520 nm for HKSOX-1, HKperox-2, and HKOH-1r, respectively (29–31).

## Confocal laser scanning microscopy

Initially, log phase *S. aureus* ATCC 43300 was suspended to $OD_{630}$ = 0.1 after being rinsed twice with 1 × PBS. After incubation for 1 h with the addition of indicated concentrations of TAF, the supernatant was removed by centrifugation, and the sediment was washed and resuspended with PBS. SYTOX Green (2 µM) was then added and incubated for 15 min in the dark, and the mixture was centrifuged and washed with PBS to remove any residual background fluorescence. Finally, the suspension was dropped on a slide and observed with a CLSM (ZeissLSM800, Jena, Germany) (32).

## Laurdan staining

MRSA ATCC 43300 was overnight cultures and diluted 1:1,000 with TSB. The bacteria were incubated at 37°C until $OD_{630}$ = 1.0, and the bacterial suspension was co-incubated with Laurdan (10 µM) for 10 min in the dark. Then, the stained bacterial suspension was washed twice with 1 × PBS and mixed with equal volumes of TAF. After 30 min at room temperature, the fluorescence intensity of Laurdan was determined with λex/λem = 350/435 nm and 350/490 nm, respectively. Membrane fluidity was quantified by the Laurdan generalized polarization (GP) index according to the following equation (33):

$$GP = \frac{A_{435nm} - A_{490nm}}{A_{435nm} + A_{490nm}}$$

## Metabolic activity determination

The metabolic activity of the bacteria was assessed according to the method reported by Wang et al. with minor modifications (34). Briefly, *S. aureus* in the log phase was washed with 1 × PBS and diluted to 0.5 McF in the presence of serial concentrations of TAF. After incubation at 37°C 180 rpm for 1 h, 90 µL of the bacterial suspension with 10 µL

AlamarBlue was further incubated at 37°C for 2 h. Metabolic conversion of resazurin to pink reduction products was evaluated by measuring $A_{570\,nm}$.

## All-atom molecular dynamics simulations

MD simulations were performed according to our previous method by CHARMM-GUI software (35). We established a lipid bimolecular model using a dioleoyl-sn-glycero-3-phosphocholine (DOPC) and 1,2-dioleoyl-sn-glycero-3phospho-(1'-rac-glycerol) (DOPG) at a ratio of 7:3. Correspondingly, a 7:3 ratio of 1-Palmitoyl-2-oleoyl-sn- glycero-3-phosphocholine (POPC) and cholesterol was used to imitate the mammalian lipid bilayer as a control (36). In brief, structural information was downloaded from NCBI, 500 ns equilibrium simulation and MD were performed by Gromacs 2018.4 software. The simulation results were visualized by the Gromacs and VMD programs.

## Tandem mass tags (TMT) quantitative proteomics analysis

Protein preparation was performed with some modifications according to the method of Zheng et al. (37). *S. aureus* ATCC 43300 was grown to $OD_{630} = 0.2$, TAF with a final concentration of $1/2 \times$ MIC was added and the mixture was incubated at 37°C and 180 rpm for 2 h. DMSO (1%, vol/vol) was used as a negative control. The sediment was then washed 3 times with $1 \times$ PBS and placed into liquid nitrogen for 15 min and subsequently stored at −80°C.

Proteins were extracted according to the procedure of Liu et al. (38). Briefly, bacteria were lysed with SDT lysis buffer (4% SDS, 1 mM DTT, 100 mM Tris-HCl, pH 7.6), and proteins were digested by trypsin and quantified with the BCA Protein Assay Kit (Bio-Rad, USA). The digested peptides were desalted with a C18 Extraction Disk (3 M Empore) and enriched by vacuum centrifugation and reconstituted in formic acid. TMT mass spectrometry analysis was performed on Thermo Q Exactivetm HF-x. Protein sequence information was obtained from the UniProt database, and raw data were analyzed using Proteome Discoverer/Spectronaut. Using R software, differentially expressed proteins (DEPs) were considered as a *P* value of < 0.05 and |fold change| of ≥1.5. Gene Ontology (GO), InterPro (IPR), and subcellular were analyzed using GSEA (gsea-3.0) enrichment (39). All the raw data and the corresponding txt files have been deposited in ProteomeXchange *via* the iProX partner repository (https://www.iprox.cn//page/project.html?id=IPX0006384000) with a data set identifier of IPX0006384000.

## Mouse subcutaneous abscess model

All animal-related procedures were approved by the Ethics Committee of the Third Xiangya Hospital of Central South University (NO. CSU-2022–0599). ICR female mice with 6–8 weeks and 23–27 g were purchased from Hunan SJA Laboratory Animal Co. The back hair of the mice was shaved using a chemical hair removal agent. Log phase cultures of *S. aureus* ATCC 43300 were washed with sterile saline and resuspended. One hundred microliters of bacterial suspension containing $1 \times 10^7$ CFU was injected subcutaneously into the back. After 1 h of infection, mice were randomly divided into three groups (*n* = 6) and were respectively treated with 20 mg/kg TAF, 30 mg/kg TAF, and 1% (vol/vol) DMSO (vehicle group) by subcutaneous injection. After 24 h of treatment, the mice were euthanized and the infected skin was excised. The tissues were homogenized with 1 mL saline and then performed CFU count (28). Meanwhile, the skin was fixed in 4% paraformaldehyde for up to 24 h for histological analysis. After dehydration with a series of alcohol concentrations and replacement with xylene, the tissue was paraffin-embedded and sectioned. After dewaxing, separate hematoxylin-eosin (H&E) and Masson's trichrome staining were performed. In addition, according to the reagent manufacturer's instructions (Servicebio, Wuhan, China), interleukin (IL)-6 and tumor necrosis factor-alpha (TNF-α) were performed and observed under an upright fluorescence microscope (Nikon E100, Japan) (40).

## *In vivo* toxicity

The *in vivo* toxicity assay was performed using 6–8 weeks of female ICR mice with intraperitoneal injection. For the time-survival assay, mice were randomly divided into four groups: vehicle group, 100 mg/kg, 50 mg/kg, and 40 mg/kg of TAF. The survival and emotional state were tracked daily for 7 days following the injection. For in-depth *in vivo* toxicity assessment, mice were randomly divided into a vehicle group and 30 mg/kg of the TAF-treated group, after 24-h treatment, whole blood and serum were collected for hematological analysis (BC-5000vet hematology analyzer, Mindray, Shenzhen, China) and biochemical marker determination (Labospect 003 automatic biochemical analyzer, Hitachi, Japan). Finally, the heart, liver, spleen, lungs, and kidneys were excised and stained for H&E observation (28).

## Statistical analysis

The experiments were repeated three times independently. Data were analyzed using the GraphPad Prism 8.0 and presented as mean ± standard deviation (SD). The Student's *t*-test was used to calculate *P* values for comparisons between two groups, and one-way ANOVA was used to analyze data between multiple groups. $P < 0.05$ was defined as statistically significant.

## RESULTS

### Bactericidal activity of TAF against *S. aureus*

We carried out high-throughput screening using the MRSA-type strain USA300 to explore promising antibacterial molecules. The workflow of high-throughput screening is shown in Fig. 1A. Based on the results of three-round screening, we finally selected TAF for an in-depth study. The molecular structure formula of TAF is shown in Fig. 1B. TAF exhibited effective bactericidal activity against type strains of both MRSA and MSSA with MICs and MBCs of 8 µg/mL and 16–64 µg/mL, respectively. TAF also showed similar bactericidal activity against *S. aureus* clinical isolates with MICs and MBCs of 8 µg/mL and 16–32 µg/mL, respectively. In addition, TAF exhibited a bactericidal effect against *S. epidermidis* and Enterococcus with MICs of 4–8 µg/mL, and MBCs of 8–64 µg/mL. However, no antibacterial activity against Gram-negative bacteria was observed with MICs more than 64 µg/mL (Table 1).

Furthermore, the bactericidal kinetics of TAF were assessed. As shown in Fig. 2A, even sub-MIC (1/2 × MIC) of TAF exhibited an effective growth-inhibiting effect against *S. aureus* within 24-h incubation. 1 × MIC of TAF could significantly reduce the bacterial counts to the limit of detection within 8-h treatment (Fig. 2B). And TAF equal to or greater than 2 × MIC killed bacteria completely within 4 h in a dose-dependent manner

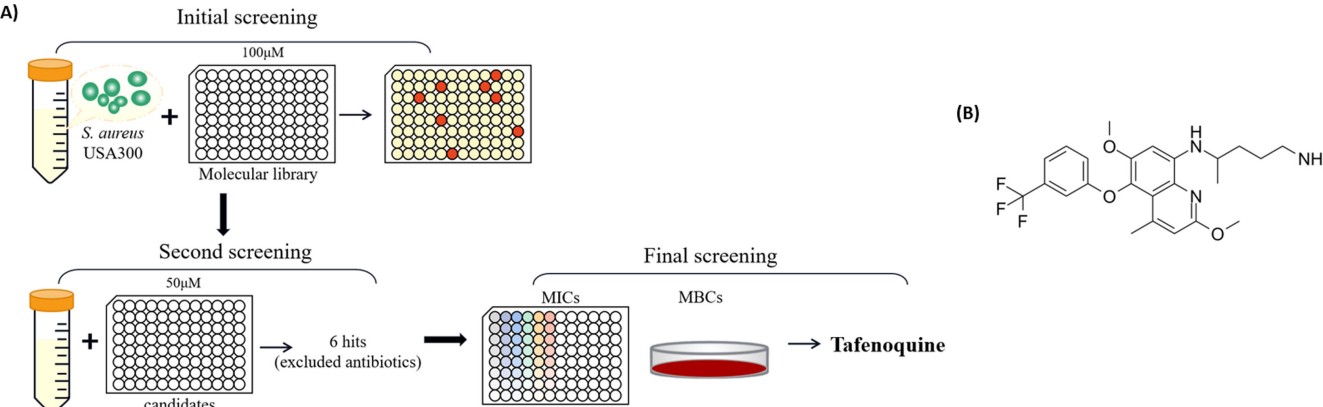

**FIG 1** Antimicrobial compound screening. (A) High-throughput screening workflow of the FDA-approved molecular library. (B) The chemical structure formula of TAF.

**TABLE 1** Antimicrobial susceptibility test of TAF against pathogens

| Strains | MIC (µg/mL) | MBC (µg/mL) |
|---|---|---|
| *S. aureus* | | |
| ATCC 43300[a] | 8 | 64 |
| USA300[a] | 8 | 32 |
| ATCC 29213 | 8 | 32 |
| ATCC 25923 | 8 | 16 |
| Newman | 8 | 32 |
| LZB1 | 8 | 16 |
| SA1191 | 16 | 16 |
| SA1192 | 8 | 16 |
| SA1193 | 8 | 32 |
| SA1195 | 8 | 16 |
| SA0645[a] | 16 | 16 |
| SA2174[a] | 16 | 16 |
| *S. epidermidis* | | |
| ATCC 12228 | 4 | 64 |
| RP62A | 4 | 64 |
| *E. faecalis* | | |
| ATCC 19212 | 8 | 8 |
| *E. faecium* | | |
| ATCC 19434 | 8 | 8 |
| *K. pneumoniae* | | |
| ATCC 700603 | >64 | >64 |
| LH2020 | >64 | >64 |
| *P. aeruginosa* | | |
| PAO1 | >64 | >64 |
| *E. coli* | | |
| ATCC 25922 | >64 | >64 |

[a]MRSA.

(Fig. 2B). By resistance inducing assay, sub-MIC of conventional antibiotics of CIP and RFP revealed a 16- to 32- and 128-fold increase in MICs within 15 days of treatment against *S. aureus* ATCC 43300, USA300, and ATCC 25923, respectively. However, the MICs of TAF still remained at the initial value during the entire process (Fig. 2C). In addition, the induced CIP- and RFP-resistant *S. aureus* also exhibited susceptibility to TAF with MICs of 8–16 µg/mL (Fig. 2D). Next, the checkerboard assay was used to evaluate the combinational antibacterial activity between TAF and conventional antibiotics. A partial synergistic antimicrobial effect was observed between TAF and gentamicin with FICI of 0.75. However, there were no beneficial synergistic effects found when TAF was combined with other conventional antibiotics such as tetracycline, ampicillin, and cefazolin (Table 2).

## Antimicrobial activity of TAF against *S. aureus* high-resistant phenotypes

Persister cells were constructed as previously reported by Kim et al. (36). Although the persister cells were highly tolerant to conventional antibiotics of VAN, DAP, and CIP (Fig. 3A), TAF could kill stationary-phased persister cells of MRSA in a dose- and time-dependent manner. For example, 8 × MIC of TAF could reduce the CFU counts of persister cells (including ATCC 43300 and USA300) from 8 to 5–6 Log10 CFU/mL after 6-h incubation (Fig. 3A). Furthermore, by using CV staining, the anti-biofilm effects of TAF were measured. TAF significantly inhibits the biofilm formation of MRSA ATCC 43300 at the concentration of 1 × MIC. It could completely inhibit the biofilm formation at a concentration equal to or greater than 2 × MIC (Fig. 3B). In addition, 1/2 × MIC of TAF showed biofilm-eradicating effects against the preformed biofilms of MRSA. The biofilm biomass was dramatically reduced in the presence of 16 × MIC TAF (Fig. 3B). Meanwhile,

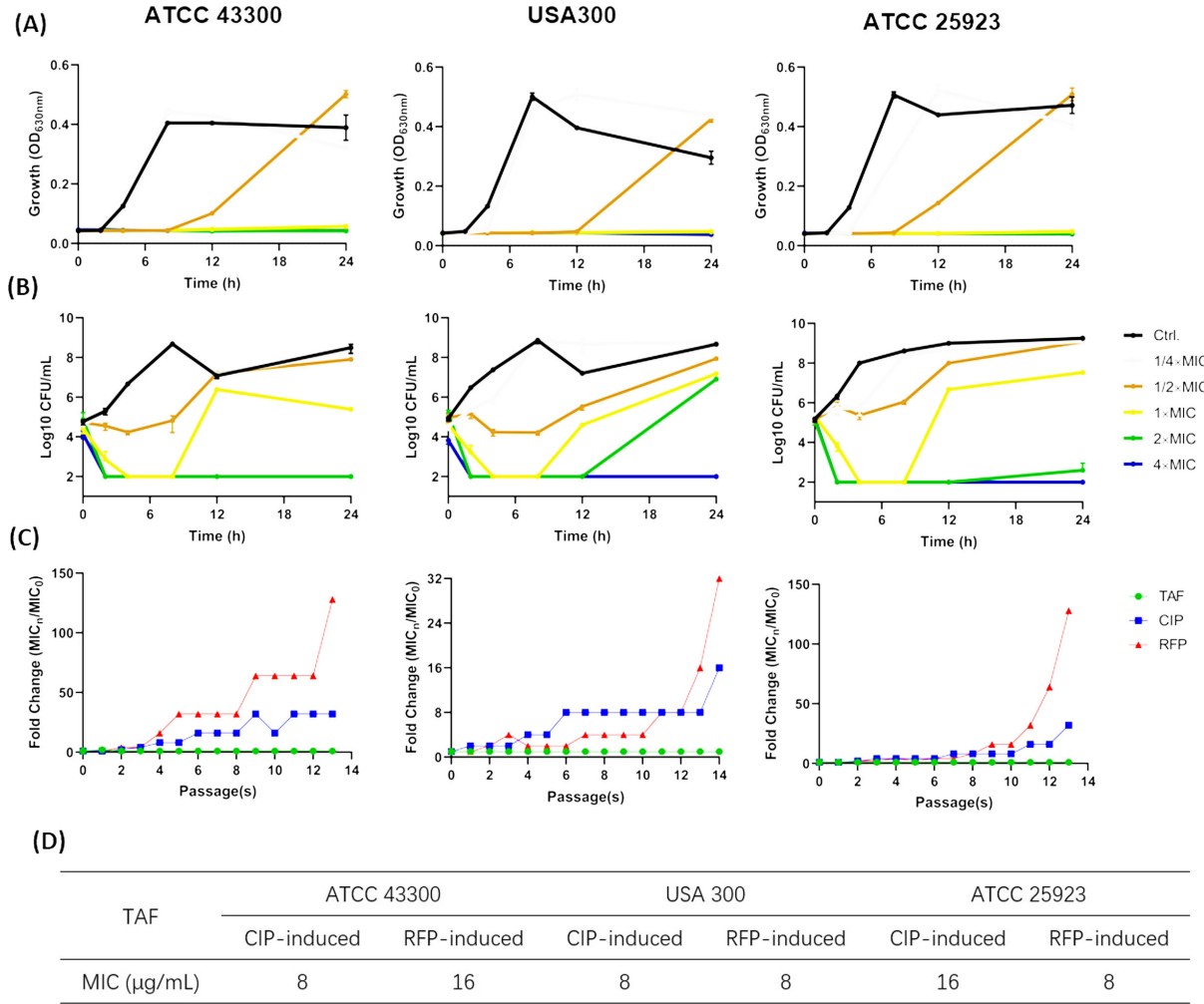

**FIG 2** Bactericidal dynamics and resistance inducing ability by TAF. (A) Time-growth curves and (B) Time-killing curves of *S. aureus* ATCC 43300, USA 300, and ATCC 25923 in the presence of 1/4 ~ 4 × MIC of TAF. 1% DMSO was used as a control. The results were presented as mean ± SD. (C) Serial passage resistance development in the presence of TAF at sub-MIC concentrations. Sub-MIC of ciprofoxacin (CIP) and rifampin (RFP) were used as controls. (D) MICs determination of TAF against CIP-/RFP-induced highly resistant *S. aureus*.

the results of planktonic cell counting showed that the method of establishing biofilm was reasonable, and TAF could effectively inhibit the number of viable bacteria while biofilm formation, and also kill the planktonic cells on the preformed biofilm (Fig. 3C)

## Antibacterial mechanism of TAF

After 1-h treatment with TAF, the EM images of *S. aureus* demonstrated obvious differences. TEM images indicated that TAF-treated cells showed mesosome-like structures with cell membrane rupture and intracellular contents leaked, whereas the control group cells showed clear and intact structures. Similarly, the SEM images exhibited varying degrees of membrane crinkling and collapse after TAF treatment (Fig. 4A). Based on these morphological changes, we hypothesized that the cell envelope might be the target of TAF. After the addition of peptidoglycan, a main cell wall component, the MIC of TAF against *S. aureus* was not altered, indicating that the cell wall was not targeted by TAF (Table S1). As we expected, the fluorescence intensity of the membrane-specific probe SYTOX Green was increased in a concentration-dependent manner within 30 min in the presence of TAF as the cell membrane disruptor melittin (Fig. 4B). Similarly, the CLSM observation showed the TAF-treated bacterial cells were

**TABLE 2** Combinational antimicrobial effects between TAF and conventional antibiotics against *S. aureus*

| Groups | Drugs[a] | MIC$_{alone}$ | MIC$_{combined}$ | Fold change | FICI | Outcome |
|---|---|---|---|---|---|---|
| 1 | TAF | 8 | 4 | 0.5 | 1 | Addition |
|   | DOX | 0.25 | 0.125 | 0.5 | | |
| 2 | TAF | 8 | 8 | 1 | 2 | Indifference |
|   | TOB | 128 | 128 | 1 | | |
| 3 | TAF | 8 | 4 | 0.5 | 1 | Addition |
|   | TET | 2 | 1 | 0.5 | | |
| 4 | TAF | 8 | 4 | 0.5 | 0.75 | Partial synergy |
|   | GEN | 32 | 8 | 0.25 | | |
| 5 | TAF | 8 | 4 | 0.5 | 1 | Addition |
|   | AMP | 16 | 8 | 0.5 | | |
| 6 | TAF | 8 | 8 | 1 | 2 | Indifference |
|   | LZD | 1 | 1 | 1 | | |
| 7 | TAF | 8 | 4 | 0.5 | 1 | Addition |
|   | CEZ | 4 | 2 | 0.5 | | |
| 8 | TAF | 8 | 8 | 1 | 2 | Indifference |
|   | E | >256 | >256 | 1 | | |
| 9 | TAF | 8 | 8 | 1 | 2 | Indifference |
|   | CLA | >256 | >256 | 1 | | |

[a]DOX, doxycycline; TOB, tobramycin; TET, tetracycline; GEN, gentamicin; AMP, ampicillin; LZD, linezolamide; CEZ, cefazolin; E, erythromycin; CLA, clarithromycin.

largely stained with SYTOX Green (Fig. 4D). Using DiSC3(5) staining, the fluorescence intensity of TAF-treated bacteria gradually decreased in a concentration-dependent manner within 5 min, indicating that TAF polarized the cell membrane (Fig. 4C). To imitate the cell membrane disrupting by TAF, we established an MD model. As shown in Fig. 4E, TAF was uptake onto the surface of the bacterial membrane within the first 10 ns of the MD simulation, and the amino groups of TAF formed a hydrogen bond with the phosphate group in the DOPC to anchor TAF to the simulated surface. Subsequently, TAF gets closer to the centroid of the membrane and penetrates the membrane after a few hundred nanoseconds, eventually binding stably in the outer leaflet of the phospholipid bilayer (Movie S1). As we expected, TAF does not interact with mammalian-mimetic lipid bilayers (Movie S2). To further investigate the affinity of TAF for phospholipid membranes and the driving force to penetrate the membranes, we calculated and analyzed the binding energies. As shown in Fig. 4F, at the beginning of the simulation, the affinity of TAF for 7DOPC/3DOPG phospholipid membranes increased rapidly, indicating that TAF can bind to bacterial membranes rapidly and then enter the interior of membranes by constant extrusion. This process is further demonstrated by reduced centroid and increased number of hydrogen bonds between TAF and 7DOPC/3DOPG phospholipid membranes. As the simulation progresses, the distance between TAF and bacterial phospholipid membrane decreases rapidly, indicating that TAF can adsorb on the surface of the phospholipid membrane quickly and basically tends to be stabilized after 100 ns, with an average value of 1.02 ± 0.23 nm after 300 ns. While in the 7POPC/3cholesterol system, TAF was not stably bound to the phospholipid membrane, with an average value of 3.84 ± 0.64 nm after 300 ns (Fig. 4H). As shown in Fig. 4I, the number of hydrogen bonding of TAF to 7DOPC/3DOPG phospholipid membranes is greater, which may be more favorable for the binding of TAF to phospholipid membranes. At the same time, the insertion of TAF caused a significant change in bilayer thickness. By contrast, the mammalian bilayer thickness was still kept unchanged (Fig. 4G). The three-dimensional binding mode revealed that in addition to the amino group of TAF forming hydrogen bonds, the aromatic ring with strong rigidity and hydrophobicity penetrates deeply into the phospholipid membrane in the 7DOPC/3DOPG phospholipid membrane system and forms a strong hydrophobic interaction with the hydrophobic bonds of phospholipid molecules, which not only stabilizes the binding but also disturbs the stability of

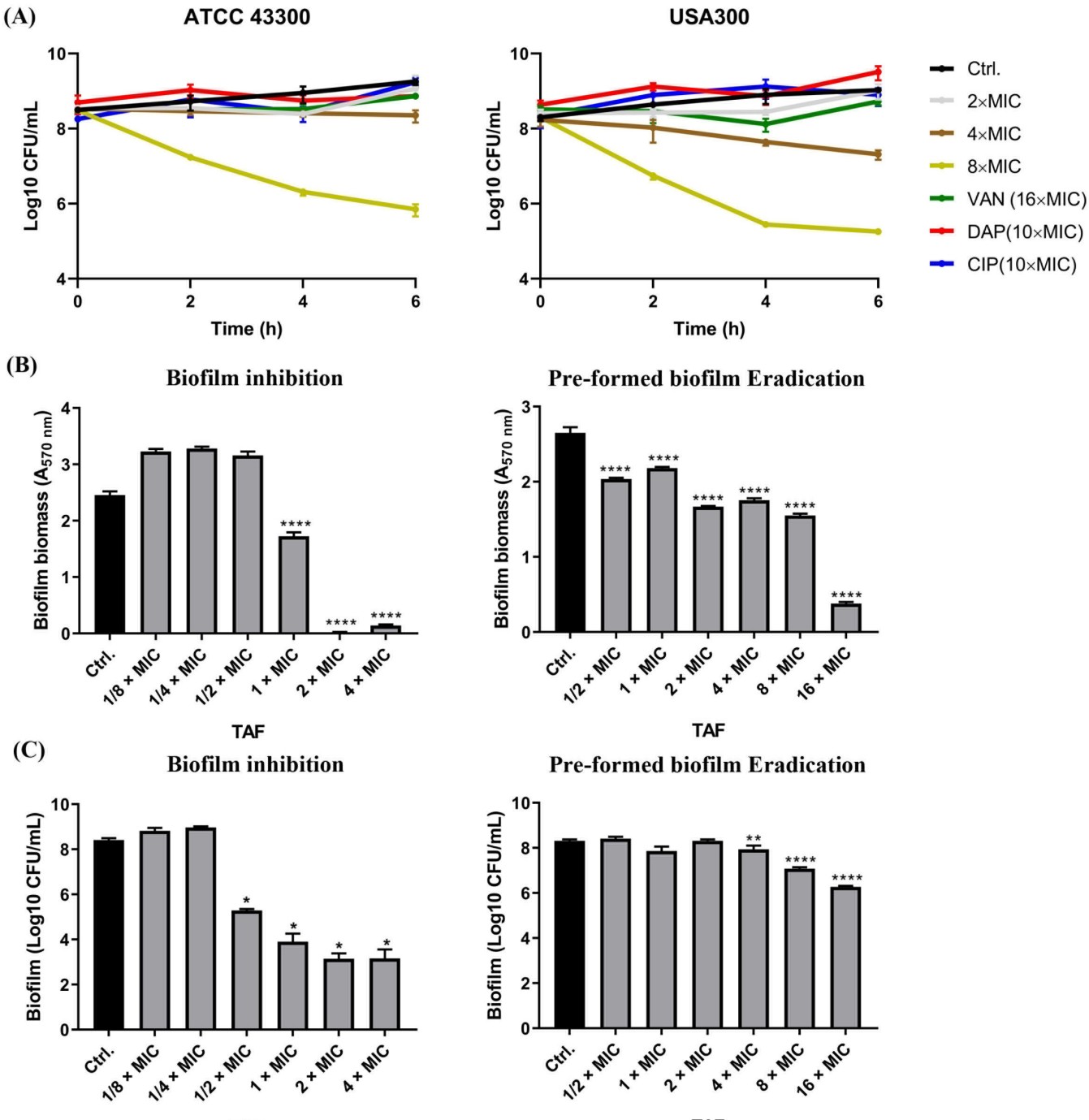

**FIG 3** Antimicrobial activity of TAF against *S. aureus* high resistant phenotypes. (A) Bactericidal effects of TAF against stationary-phased persister cells of MRSA ATCC 43300 and USA300. Vancomycin (16 × MIC), DAP (10 × MIC), and CIP (10 × MIC) were used as controls. Biofilm formation inhibitory and biofilm eradicating effects of TAF against *S. aureus* ATCC 43300 determined by crystal violet staining (B) and planktonic cell count (C). (B and C) The left column and the right column represented biofilm inhibition and eradication, respectively. Data are presented as means ± SD and analyzed by one-way ANOVA. ****$P < 0.0001$.

phospholipid molecules (Fig. 4J). Subsequently, we investigated the membrane fluidity after TAF treatment using Laurdan, a membrane-sensitive probe. The results showed that the Laurdan GP of *S. aureus* decreased gradually in a dose-dependent manner in the presence of TAF, suggesting that TAF may permeate into the lipid bilayer of the cell membrane and cause changes in membrane fluidity (Fig. 4K). Oxidative damage

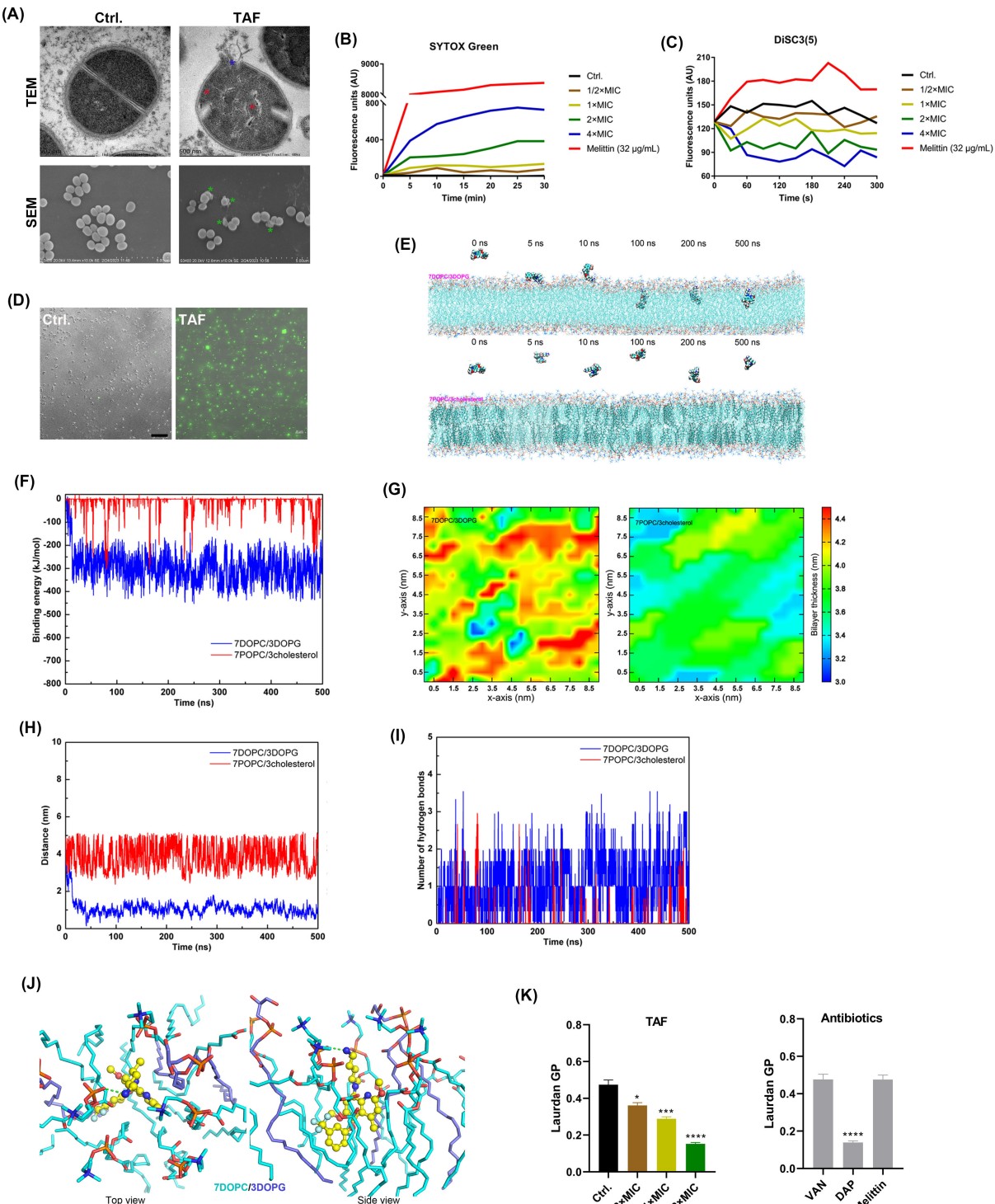

**FIG 4** Mechanism of action by TAF. (A) SEM and TEM observation of *S. aureus* after being treated with 5 × MICs of TAF for 1 h. The blue asterisk indicates cell membrane rupture and intracellular contents leakage. A red asterisk indicates mesosome-like structures. A green asterisk indicates cell membrane crinkling and collapse. Scale bars: 5 µm for SEM, 500 nm for TEM. (B) Membrane permeabilization detected by SYTOX Green. (C) Membrane potential detected by DiSC3 (5). Melittin (32 µg/mL) was used as a positive control. (D) Representative SYTOX Green staining CLSM images of *S. aureus* after being treated with TAF (1/2 × MIC) for 1 h. Scale bar: 20 µm. (E) Representative configurations of MD simulations of the bacterial (upper panel) and mammal (down panel) lipid bilayer in the presence of TAF. From left to right: the onset of simulation, membrane attachment, membrane penetration, and equilibrium state of the TAF interaction with the cell membrane. (F) Changes in binding energy of TAF interacted with lipid bilayers. (G) Lipid bilayer thickness alteration by TAF treatment. (H) Changes of distance to the bacterial or mammal lipid bilayers centroid by TAF. (I) Hydrogen bond alterations between TAF and cell membranes. (J) Top and side views of simulation

**FIG 4** (Continued)

of the interaction between TAF and adjacent lipids by molecular docking. Cyan lines indicate phospholipid membrane structures, yellow ball-and-stick models indicate TAF, and green dashed lines indicate hydrogen bonding interactions. (K) Cell membrane fluidity was detected by Laurdan staining. Melittin (32 μg/mL), VAN, and DAP were used as controls. All experiments were performed using MRSA ATCC 43300. Data are analyzed by one-way ANOVA. Error bars represent means ± SD. *$P < 0.05$, **$P < 0.01$, ***$P < 0.001$, ****$P < 0.0001$.

usually disrupts the bacterial structure and inhibits intracellular protein activity, which may lead to bacterial death (41). In this study, by using probe H2DCFDA, we found TAF-enhanced ROS levels of bacteria in a dose- and time-dependent manner (Fig. 5A). Subsequently, the specific components of ROS were determined, including superoxide anion ($O_2 \cdot -$), hydrogen peroxide ($H_2O_2$), and hydroxyl radicals ($\cdot OH$). As shown in Fig. 5B and D, the levels of $H_2O_2$ and $\cdot OH$ significantly increased after TAF treatment. In addition, the antibacterial activity of TAF was also reduced with the MIC of 32 μg/mL in the presence of exogenous glutathione (a ROS scavenger) (Table S1). Subsequently, by using AlamarBlue staining, the metabolic activity of *S. aureus* was also found to be significantly hindered by TAF (Fig. 5F).

Furthermore, proteomic changes in *S. aureus* were analyzed after exposure to TAF. The subcellular localization of DEPs exhibited 16.67% of the DEPs were found in cytoplasm protein, 26.19% in cell membrane protein, and 57.14% in extracellular protein (Fig. 6A). A total of 127 DEPs were found in TAF-treated group, 33 of which were upregulated and 94 were downregulated (Fig. 6B). Notably, a total of 27 core enrichments were identified in cell membrane protein, and the shape of the leader subset appeared in the ES chart with ES = 0.22, indicating that the TAF-treated *S. aureus* cell membrane protein was significantly enriched (Fig. 6C). In all, 21 of the DEPs were associated with cell wall/membrane/envelope biogenesis and seven with lipid transport and metabolism (Fig. 6D). In conclusion, the results of the proteomic analysis still support the membrane disrupting activity by TAF.

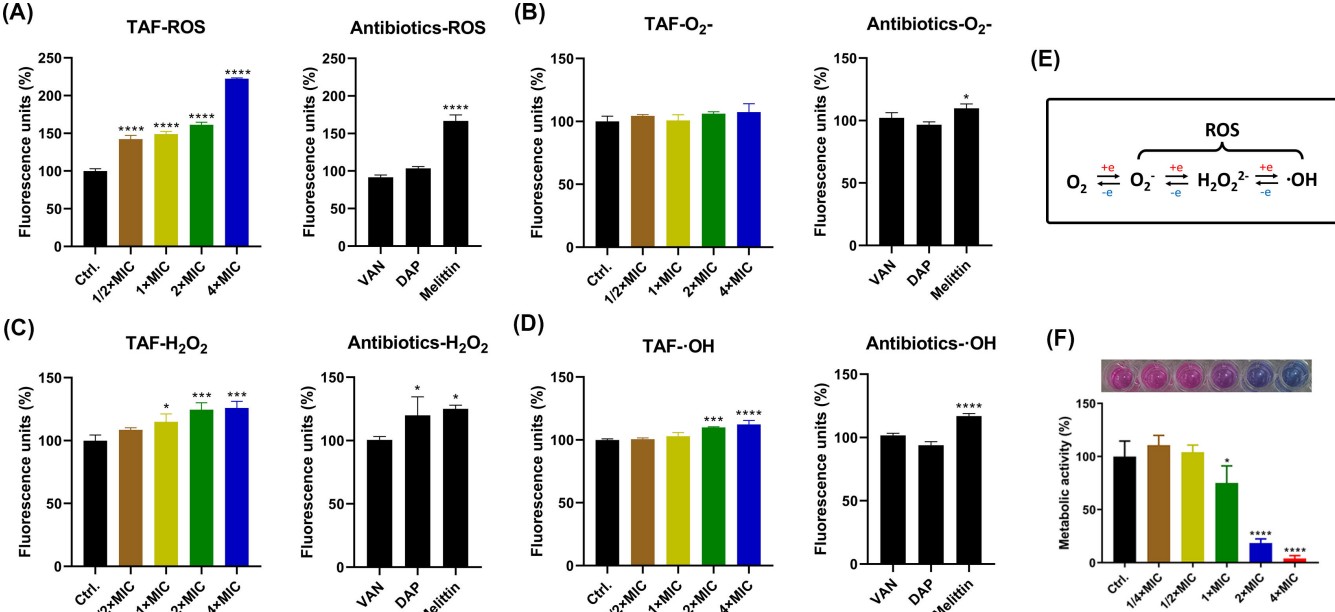

**FIG 5** Oxidative stress of TAF against *S. aureus*. (A) ROS production quantification by H2DCFDA staining. The specific reactive species of ROS, including $O_2 \cdot -$ (B) $H_2O_2$ (C) and $\cdot OH$ (D) were determined by the probes of HKSOX-1, HKperox-2, and HKOH-1r, respectively. (E) Schematic diagram of the interconversion of specific reactive species. (F) Metabolic activity determination after treatment with TAF.

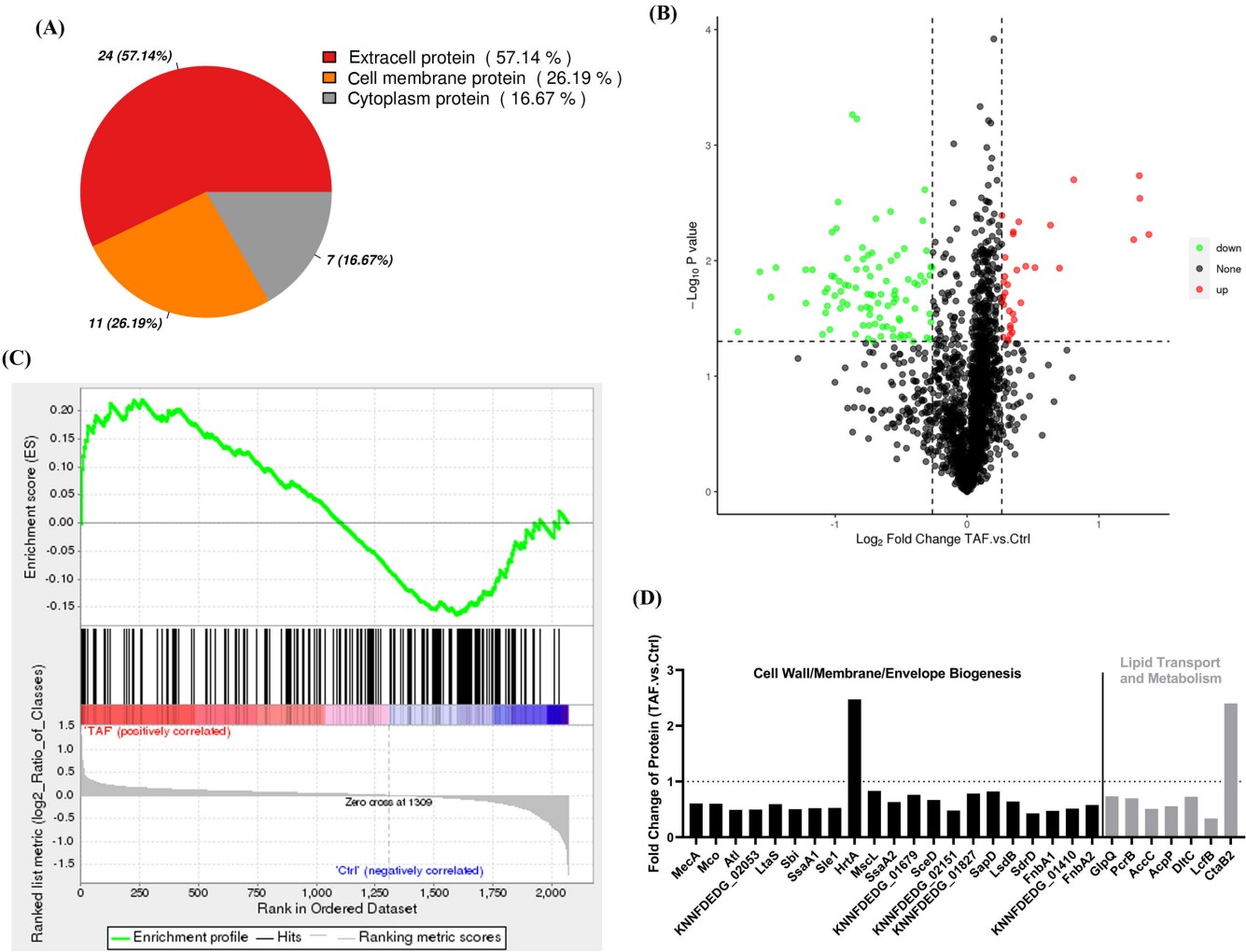

**FIG 6** Proteomics analysis. (A) Subcellular localization of DEPs. (B) Volcano map analysis of 127 DEPs, including 33 upregulated proteins (red) and 94 downregulated proteins (green). (C) GSEA enrichment analysis of cell membrane proteins. (D) Detailed information in clusters of orthologous groups (COG) analysis of membrane-associated DEPs.

## *In vivo* antimicrobial efficacy by TAF

We used a subcutaneous abscess model to assess the *in vivo* antimicrobial activity of TAF (Fig. 7A). As shown in Fig. 7B, 20 mg/kg and 30 mg/kg of TAF-treated groups significantly reduce the abscesses bacterial loads by ~1 and ~2 Log10 CFU/mL, respectively, compared with the vehicle group. Figure 7C shows the representative images of live bacteria counts. Tissue inflammatory cell infiltration, pro-inflammatory cytokine production, and tissue healing could be assessed by H&E staining and IHC, respectively. As shown in Fig. 7E, there was a large infiltration of inflammatory cells in the vehicle group, while the TAF-treated group had a significantly smaller region of inflammatory infiltration and a significantly lower number of inflammatory cells. The IHC results for IL-6 and TNF-α showed that there were fewer pro-inflammatory cytokines in the TAF-treated group compared with the vehicle group (Fig. 7F and G), indicating a diminished inflammatory response in treatment groups. In addition, Masson staining showed increased collagen fibers and promoted tissue healing in the TAF-treated group (Fig. 7H).

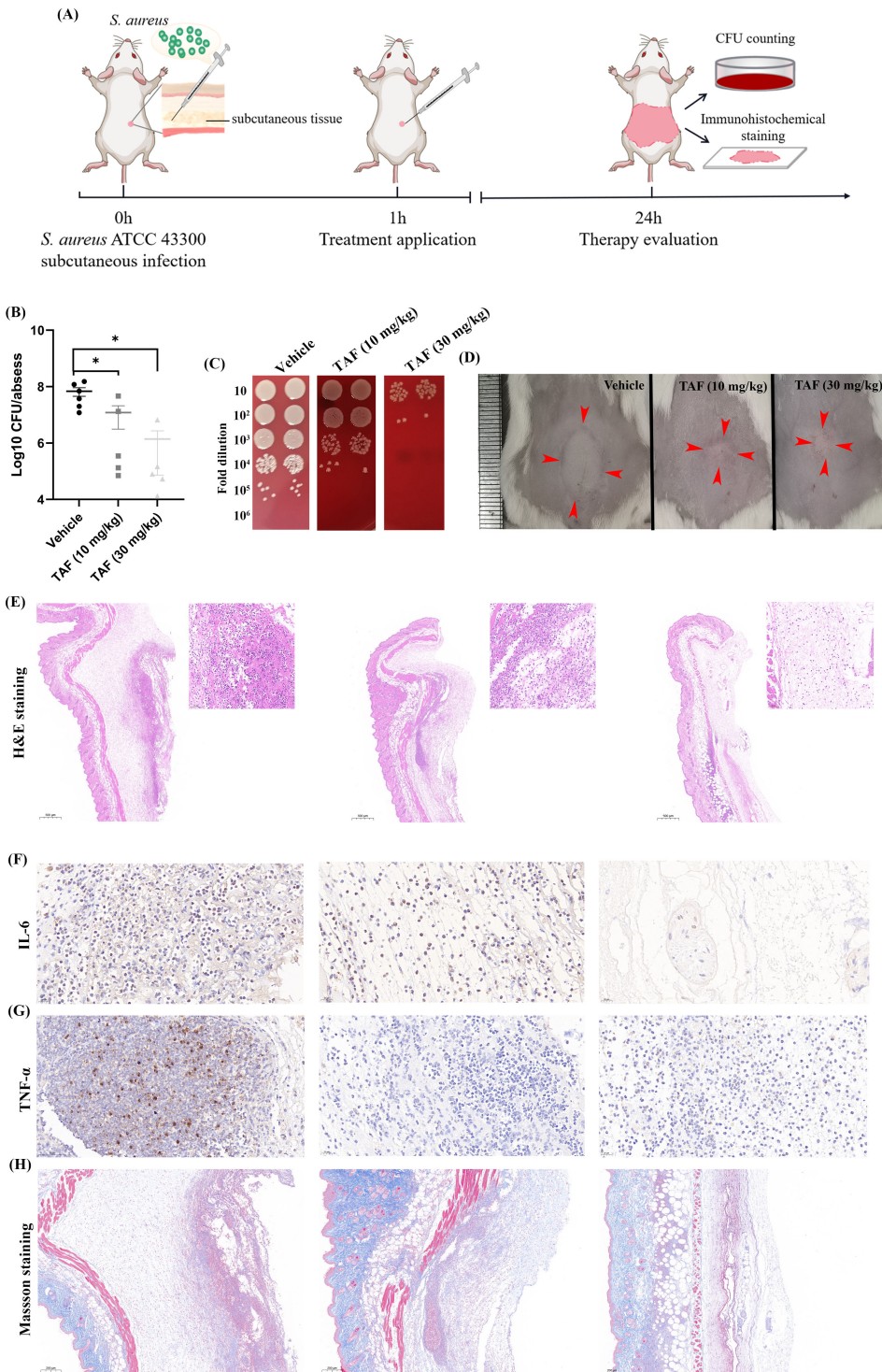

**FIG 7** *In vivo* antimicrobial efficacy of TAF in a subcutaneous abscess model. (A) Workflow of the subcutaneous abscess model. For the treatment application, mice (*n* = 6 mice/group) were subcutaneously injected with 20 mg/kg and 30 mg/kg of TAF and 1% DMSO, respectively. (B) Viable cell counts in the abscess. (C) The representative images of CFU counts. Five microliters of fold diluted bacterial suspension was dropped on the sheep blood agar. (D) The representative images of the abscess. Red arrows indicate the boundaries of the abscess. Histological analysis of the abscess after being treated with TAF by H&E (E) and Masson staining (H), and immunohistochemical analysis of IL-6 (F) and TNF-α(G), respectively. Data are presented as means ± SD and analyzed by one-way ANOVA.*P* < 0.05.

## Acceptable toxicity of TAF *in vivo*

To determine the highest tolerant dose of TAF in mice, we first performed a time-survival assay by intraperitoneal injection (i.p.). As shown in Fig. S1A, 100 mg/kg of TAF was lethal to all the tested mice within 12 h. TAF reached the half-lethal dose at 50 mg/kg while surviving mice exhibited varying degrees of mental depression and loss of appetite. TAF at 40 mg/kg did not cause death and the mice were in good spirits, which indicates the acceptable dosage *in vivo*. Therefore, the choice of 30 mg/kg for the mice infection model in our study is acceptable. Furthermore, the in-depth *in vivo* toxicity of 30 mg/kg TAF (i.p.) was evaluated. As we expected, there were no statistically significant changes between the control group and the TAF-treated group in the renal and myocardial functional biomarkers of BUN and CK, respectively (Fig. S1B and C). Similarly, the hematological parameters including white blood cell count, neutrophil percentage, platelet count, and red blood cell count also showed no statistically significant differences (Fig. S1D to G). The H&E staining revealed that both of the treated and untreated groups exhibited normal organ morphology with no significant pathological changes. In conclusion, these findings demonstrated the safety of TAF *in vivo* (Fig. S1H).

## DISCUSSION

*S. aureus* is a common human pathogen that causes skin infections and even serious invasive infections (2). Due to the rapid development of drug resistance and the difficulty in developing new antibiotics, drug repurposing has emerged as a promising alternative. In this study, the antimalarial agent TAF demonstrated potent antibacterial efficiency against *S. aureus*. Our study also found that TAF was able to kill persister cells, prevent biofilm formation, and eradicate preformed biofilms. In addition, TAF exhibited an antibacterial effect in mice *in vivo*. To the best of our knowledge, this is the first systematic study on the antibacterial effects against *S. aureus* and potential mechanisms of TAF.

Besides our study, antimalarials, except for TAF, have demonstrated diverse antimicrobial activities. For example, artemisinin and its derivatives, mefloquine, and chloroquine have been reported to exhibit antibacterial or antifungal effects (19, 42–44). A series of primaquine derivatives have been shown broad-spectrum antimicrobial activity (45). Similarly, as a well-known antibacterial agent, aminoquinoline and its derivatives have established antimalarial, antiparasitic, antibacterial, and antifungal activities (46). Therefore, the chemical backbone of aminoquinoline may be the basis for the antimicrobial action of TAF.

Mechanism study indicates that TAF is selectively targeted to the bacterial cell membrane. SYTOX Green interacts with nucleic acids through damaged cell membranes and is used to monitor alterations in bacterial cell membrane permeability (47). DiSC3(5) reflects the potential change in the cell membrane and can be used as an indicator of membrane polarization to reflect early bacterial damage (48). In our study, by using the above fluorescent probes, *S. aureus* exhibited increased permeability and membrane hyperpolarization after being treated with TAF, indicating that the bacterial cell membrane was damaged. Similar to the report by Li et al. (24), the membrane potential of TAF exhibited hyperpolarization, which was different from the positive control by melittin. Moreover, previous studies have reported that the antimicrobial agents of citral, basil, white sagebrush, and sweet acacia can also hyperpolarize the bacterial membranes (48, 49). So, both hyperpolarization and depolarization are indicative of bacterial cell membrane damage, and the hyperpolarization observed in our study could potentially be attributed to the alterations in H$^+$ levels or disruptions in the homeostasis of the cell membrane (48). The results of MD simulations and electron microscopy images supported this hypothesis. And similar to the study of Wang et al. (34), the bacterial metabolic activity was inhibited after TAF treatment. In addition, TAF promotes ROS production, which can further lead to bacterial damage (41). In summary, we hypothesized that TAF targets the *S. aureus* cell membrane, causing changes in cell membrane

permeability and proton dynamic potential, and promoting bacterial oxidative stress, which further leads to bacterial death.

The results of proteomics also supported the above hypothesis. Previous studies have shown that fatty acids, as components of phospholipids and lipopolysaccharides, are important in maintaining the integrity and permeability of bacterial cell membranes, and acyl carrier protein and acetyl-CoA carboxylase are the first steps of fatty acid biosynthesis (50). These proteins, AccP and AccC, were downregulated after TAF treatment, suggesting that TAF may inhibit fatty acid synthesis and further contribute to the disruption of cell membranes and membrane fluidity. MscL is an important channel protein in the bacterial cell membrane, which acts as an "emergency release valve," maintaining normal osmotic pressure (51, 52). Proteomics results showed that McsL was downregulated after TAF treatment, suggesting that TAF alters the homeostasis of bacterial cell membranes by inhibiting the expression of McsL. Similarly, bioinformatics analysis of some membrane-targeted agents showed similar alterations, such as cuminaldehyde, kurarinone, and sophoraflavaone G (50, 51). Moreover, some proteins, including FnbA and Atl (53, 54), that promote cell adhesion and biofilm formation were downregulated after TAF treatment, which indicates that the antibiofilm ability of TAF could be a promising option for the treatment of biofilm-related infections. In addition, other downregulated membrane proteins may also be the direct or indirect potential targets of the TAF. For example, lipoteichoic acid synthase (LtaS) is an important enzyme for cell viability (55), the lack of which often leads to a defective bacterial envelope (56), and some LtaS inhibitors often exhibit bacteriostatic effects against *S. aureus* (57). In this study, proteomic analysis provided evidence and other potential research directions about the mechanism of action of TAF, which need to be further demonstrated in future studies.

*S. aureus* biofilm often forms on the surface of medical devices and is a common cause of catheter-related or joint replacement-associated infections (5). Biofilm is composed of the extracellular matrix, and the cells within the biofilm are low metabolically active with the formation of high-resistant persister cells (10). In our study, TAF exhibited bactericidal effects on persister cells and biofilms, and the effect on the biofilm matrix was highly significant, which provides a strategy for treating biofilm-associated infections. Significantly, the planktonic cells were more resistant to TAF treatment in biofilm eradication compared to inhibition. The composition of biofilm cells is complex and dynamic (58), the persister cells and metabolically dormant cells exist within the biofilms and are protected by the biofilm matrix (59). We hypothesized that the biofilm matrix was disrupted during biofilm eradication so that the antibiotic-resistant bacteria were dispatched from the biofilm and became planktonic cells, which showed resistance to TAF. In addition, one advantage of membrane-active antibiotics is that they are more readily to kill persister cells. Because these drugs disrupt the bacterial cell membrane independently of the metabolic state (14). Similarly, some antibacterial agents that target membranes also exhibit activity against persister cells and biofilm, such as the small molecule SCH-79797, and the antimicrobial peptide SAAP-148 (60, 61).

TAF is an oral antimalarial drug often with a single dose of 300 mg administered for patients. A clinical study showed that subjects receiving 400 mg TAF daily for 3 days and then 400 mg TAF monthly for 5 months had a well-tolerated and plasma clearance half-life of 14 days (62). A study by Vuong et al. showed that 20 mg/kg TAF administered orally for 14 days was safe in mice (63). Similarly, in our study, administered 30 mg/kg TAF (i.p.) had no effect on the survival of mice, and no hemolysis or tissue toxicity was observed. Antibacterial drugs targeting the bacterial cell membrane carry the risk of targeting host cell membranes, but the application of daptomycin and oritavancin illustrates the possibility of bacterial selectivity (14). Compared to Gram-positive bacteria, mammalian cell membranes have more cholesterol that protects cells from antimicrobial agents (36, 64), and MD simulation results also showed that TAF does not bind to mammalian lipid bilayers without reduction in cell membrane thickness. In our study, no significant changes in erythrocytes, hemoglobin, and platelets were observed in mice

in the short-term treatment, suggesting that TAF has the potential to be developed as an antimicrobial agent. Undeniably, studies on structural optimization, changing the delivery mode, new delivery systems, and adding detoxification adjuvants are needed to reduce the toxicity of TAF and improve its bioavailability (65–68).

In conclusion, TAF, as a selective membrane-active agent, exhibits strong antibacterial activity against planktonic, biofilm, and persister cells of MRSA. TAF did not cause drug resistance and has a good capacity to treat infections caused by MRSA with acceptable toxicity *in vivo*. These results suggest that TAF provides a promising approach to treat infections caused by MRSA planktonic cells, biofilms, or persister cells.

## ACKNOWLEDGMENTS

This study was supported by the National Natural Science Foundation of China (grant numbers 82072350 and 82202591), the Natural Science Foundation of Hunan Province (grant number 2022JJ70046), the Key Research and Development Program of Hunan Province of China (grant number 2022SK2116), and the Project of Scientific Research Plan of Hunan Provincial Health Commission (grant no: B202311000022).

P.S. and Y.W. designed the research, and revised and edited the manuscript. Y.Y. and P.S. completed most of the experiments and wrote the manuscript. L.L. and L.Z. completed the experiments and data collection. P.S., Y.Y., S.L., Z.L., and Y.L. performed the data analysis. All authors contributed to the article.

## AUTHOR AFFILIATIONS

[1]Department of Laboratory Medicine, The Third Xiangya Hospital of Central South University, Changsha, China

[2]Department of Laboratory Medicine, The Affiliated Changsha Hospital of Xiangya School of Medicine (The First Hospital of Changsha), Central South University, Changsha, China

## AUTHOR ORCIDs

Yong Wu  http://orcid.org/0000-0002-3667-8716

## FUNDING

| Funder | Grant(s) | Author(s) |
| --- | --- | --- |
| MOST \| National Natural Science Foundation of China (NSFC) | 82072350, 82202591 | Yong Wu |
| HSTD \| Natural Science Foundation of Hunan Province (湖南省自然科学基金) | 2022JJ70046 | Yong Wu |
| Key Research and Development Program of Hunan Province of China (湖南省重点研发计划) | 2022SK2116 | Yong Wu |
| Project of Scientific Research Plan of Hunan Provincial Health Commission | B202311000022 | Pengfei She |

## AUTHOR CONTRIBUTIONS

Pengfei She, Data curation, Methodology, Visualization, Writing – original draft, Writing – review and editing | Yifan Yang, Formal analysis, Investigation, Methodology, Visualization, Writing – original draft | Linhui Li, Formal analysis, Investigation, Methodology, Visualization | Yimin Li, Software, Visualization | Shasha Liu, Software, Visualization | Zehao Li, Software, Visualization | Linying Zhou, Software, Visualization | Yong Wu, Conceptualization, Project administration, Resources, Supervision, Writing – review and editing

## DATA AVAILABILITY

All the raw data and the corresponding text files have been deposited in ProteomeX-change *via* the iProX partner repository with a data set identifier of IPX0006384000.

## ETHICS APPROVAL

The animal study was reviewed and approved by the Ethics Committee of the Third Xiangya Hospital of Central South University (NO. CSU-2022-0599).

## ADDITIONAL FILES

The following material is available online.

### Supplemental Material

**Supplemental material (mSystems01026-23-s0001.docx).** Table S1, Figure S1, and supplemental movie legends.
**Movie S1 (mSystems01026-23-s0002.mp4).** MD simulation process of TAF interaction with 7DOPC/3DOPG phospholipid membranes. TAF rapidly entered and stably bound in the outer leaflet of the phospholipid bilayer.
**Movie S2 (mSystems01026-23-s0003.mp4).** MD simulation process of TAF interaction with 7POPC/3cholesterol phospholipid bilayer. TAF did not bind with 7POPC/3choles-terol phospholipid bilayer.

### Open Peer Review

**PEER REVIEW HISTORY (review-history.pdf).** An accounting of the reviewer comments and feedback.

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
