## [Reviewer comments · mSystems]

Repurposing of the anti-malarial agent tafenoquine to combat MRSA

Pengfei She, Yifan Yang, Linhui Li, Yimin Li, Shasha Liu, Zehao Li, Linying Zhou, and Yong Wu

Corresponding Author(s): Yong Wu, The Affiliated Changsha Hospital of Xiangya School of Medicine, Central South University

Review Timeline:

Submission Date:	September 25, 2023
Editorial Decision:	October 11, 2023
Revision Received:	October 19, 2023
Editorial Decision:	October 24, 2023
Revision Received:	October 25, 2023
Accepted:	October 25, 2023

Editor: Anupama Khare

Reviewer(s): The reviewers have opted to remain anonymous.

Transaction Report:

DOI: <https://doi.org/10.1128/msystems.01026-23>

October 11, 2023

Prof. Yong Wu
The Affiliated Changsha Hospital of Xiangya School of Medicine, Central South University
No.311, Yingpan Road
Changsha
China

Re: mSystems01026-23 (**Repurposing of the anti-malarial agent tafenoquine to combat MRSA**)

Dear Prof. Yong Wu:

Thank you for submitting your manuscript to mSystems. We have completed our review and I am pleased to inform you that, in principle, we expect to accept it for publication in mSystems. However, acceptance will not be final until you have adequately addressed the reviewer comments.

Additionally, your proteomics data on ProteomeXchange doesn't appear to be publicly available yet, so please clarify when this will be available.

Preparing Revision Guidelines

Please return the manuscript within 60 days; if you cannot complete the modification within this time period, please contact me. If you do not wish to modify the manuscript and prefer to submit it to another journal, please notify me of your decision immediately so that the manuscript may be formally withdrawn from consideration by mSystems.

Sincerely,

Anupama Khare

Editor, mSystems

Journals Department
Reviewer comments:

Reviewer #1 (Comments for the Author):

I applaud the authors for their efforts to address my concerns and in providing an improved revised manuscript. The inclusion of the Laurdan staining with appropriate controls strengthens the claim that TAF acts as a membrane disruptor. The authors also provide citations to justify their choice of lipids in the MD simulations consistent with previously published reports. Even though I remain skeptical that a mammalian lipid bilayer can be adequately modeled with two lipids, I acknowledge that this is accepted in the field.

I appreciate the authors' clarification on the different experimental setups for analyses by EM and the DiSC3(5) assays. It is reasonable that TAF treatment could lead to mild hyperpolarization in the short term (5 minutes) and membrane ruptures in the long term (1 hour). I would advise the authors to clarify these differences in the text for the readers.

Minor: Fig. 6A - It is still not clear what the distinction is between "Cell inner membrane protein" and "Cell outer membrane protein" as *S. aureus* possesses a single cell membrane. Are these DEPs that are localized to one side or the other of the membrane? Please clarify in the text for the readers.

Reviewer #2 (Comments for the Author):

Overall, the authors have addressed most of my previous comments. I still suggest to proof-read for grammar prior to publication and would like to see the following comments/concerns addressed:

1) Fig. 3A: The persister cell population in stationary phase cultures are increased compared to exponentially growing cells, however, they would not be 100%. One would expect that VAN/DAP/CIP kill at least the susceptible cells of the stationary phase culture which is not the case here. I find this quite odd. How do the authors explain that 100% of their culture appears to be antibiotic tolerant?

2) Fig 3B-C: please either give each panel a title so that the reader better understand that the left column illustrates biofilm inhibition and the right column represents eradication. Alternatively, split the figure into B-E. What how do the authors explain that planktonic cells were so much more resistant to the treatment when grown under eradication conditions as compared to biofilm inhibition?

3) I feel that the description and evaluation of the proteomic analysis, also in light of the scope of mSystems, falls short and would encourage the authors to elaborate on their findings.

Dear Reviewers:

We really appreciate you for your carefulness and conscientiousness. Your suggestions are really helpful for revising and improving our paper. According to your suggestions, we have made the following revisions (Changes in the manuscript were marked in blue) in this manuscript:

Responds to the reviewers' comments:**Reviewer #1 (Comments for the Author):**

I applaud the authors for their efforts to address my concerns and in providing an improved revised manuscript. The inclusion of the Laurdan staining with appropriate controls strengthens the claim that TAF acts as a membrane disruptor. The authors also provide citations to justify their choice of lipids in the MD simulations consistent with previously published reports. Even though I remain skeptical that a mammalian lipid bilayer can be adequately modeled with two lipids, I acknowledge that this is accepted in the field.

Comment: I appreciate the authors' clarification on the different experimental setups for analyses by EM and the DiSC3(5) assays. It is reasonable that TAF treatment could lead to mild hyperpolarization in the short term (5 minutes) and membrane ruptures in the long term (1 hour). I would advise the authors to clarify these differences in the text for the readers.

Response: Thank you for recognizing and suggesting our paper, and we have added additional descriptions about the differences in the treatment time in "Results" (lines 335 and 347).

Comment: Minor: Fig. 6A - It is still not clear what the distinction is between "Cell inner membrane protein" and "Cell outer membrane protein" as *S. aureus* possesses a single cell membrane. Are these DEPs that are localized to one side or the other of the membrane? Please clarify in the text for the readers.

Response: Thank you for your reminder! We were really sorry for our carelessness. The results of the subcellular localization were blasted by using an independent database, but the incorrect expression "cell outer membrane" was appeared due to the comparison to wrong database. This error has been corrected, and we have re-analyzed the subcellular localization data and double-checked the other data, all of which are now reliable and accurate. The relevant descriptions in "Result" were also corrected in Lines 390-391. The re-described results do not influence the conclusion as we previously made. We apologize again for our carelessness!

Reviewer #2 (Comments for the Author):

Overall, the authors have addressed most of my previous comments. I still suggest to proof-read for grammar prior to publication and would like to see the following comments/concerns addressed:

Response: Thank you for recognizing our revision! We have improved the English in the secondly revised manuscript.

Comment: 1) Fig. 3A: The persister cell population in stationary phase cultures are increased compared to exponentially growing cells, however, they would not be 100%. One would expect that VAN/DAP/CIP kill at least the susceptible cells of the stationary phase culture which is not the case here. I find this quite odd. How do the authors explain that 100% of their culture appears to be antibiotic tolerant?

Response: Thank you for your comment! The ratio of existed persister cells in stationary phase is perhaps related to the bacterial species. For example, in *E. coli*, the stationary phase cultures contain susceptible non-persister cells, so obtaining persister cells requires the addition of extra antibiotics to kill them (1, 2). However, for *S. aureus*, approximately all of the cells in stationary phase are persistent (1, 3). In a number of previously reported studies, the persister cells killing assays using the same method have also demonstrated highly tolerant to conventional antibiotics, which is similar to our results. For example, as the results reported by Jia et al. (Fig. 1A), Kim et al. (Fig. 1B), and Kim et al. (Fig. 1C), respectively, as follows:

Fig. 1A

Fig. 1B

Fig. 1C

Figure 1. Persister killing activity by conventional antibiotics.

Reference

1. Allison KR, Brynildsen MP, Collins JJ. 2011. Metabolite-enabled eradication of bacterial persisters by aminoglycosides. *Nature* 473:216-20.
2. Conlon BP, Rowe SE, Gandt AB, Nuxoll AS, Donegan NP, Zalis EA, Clair G, Adkins JN, Cheung AL, Lewis K. 2016. Persister formation in *Staphylococcus aureus* is associated with ATP depletion. *Nat Microbiol* 1.
3. Keren I, Kaldalu N, Spoering A, Wang Y, Lewis K. 2004. Persister cells and tolerance to antimicrobials. *FEMS Microbiol Lett* 230:13-8.

Comment: 2) Fig 3B-C: please either give each panel a title so that the reader better understand that the left column illustrates biofilm inhibition and the right column represents eradication. Alternatively, split the figure into B-E. What how do the authors explain that planktonic cells were so much more resistant to the treatment when grown under eradication conditions as compared to biofilm inhibition?

Response: Thank you for your suggestions! We have added the following description in the Figure legend of Figure 3: “(B-C) The left column and the right column represented the biofilm inhibition and eradication, respectively.” (lines 736-737).

The composition of cells in biofilm is complex and dynamic (1), the persister cells and metabolically dormant cells are existed within the biofilms and are protected by the biofilm matrix (2). We hypothesized that the biofilm matrix was disrupted during biofilm eradication, so that the antibiotic-resistant bacteria were dispatched from the biofilm and became planktonic cells, which showed resistant to TAF.

Reference

1. Moormeier DE, Bayles KW. 2017. *Staphylococcus aureus* biofilm: a complex developmental organism. *Mol Microbiol* 104:365-376.
2. Lister JL, Horswill AR. 2014. *Staphylococcus aureus* biofilms: recent developments in biofilm dispersal. *Front Cell Infect Microbiol* 4:178.

3) I feel that the description and evaluation of the proteomic analysis, also in light of the scope of mSystems, falls short and would encourage the authors to elaborate on their findings.

Response: We agree with your suggestions, and we have added more description of the

proteomics results in the "Discussion" part (lines 473-479).

We sincerely thank the reviewers for their valuable feedback that we have used to improve the quality of our manuscript. Changes in the manuscript were marked in blue. We hope the correction will meet with approval. Once again, thank you very much for your comments and suggestions.

Sincerely,

Prof. Wu Yong

E-mail: wuyong_zn@csu.edu.cn

Re: mSystems01026-23R1 (**Repurposing of the anti-malarial agent tafenoquine to combat MRSA**)

Dear Prof. Yong Wu:

Thank you for the privilege of reviewing your work. Below you will find my comments, and instructions from the mSystems editorial office.

1) Comment 2 from Reviewer 2 suggested that the panel in Figure 3B-C itself should be labeled. So in the figure itself, for the left column, please add a label 'Inhibition of biofilm formation' and for the right column 'Eradication of pre-formed biofilms'.

2) Please include your response to this comment (your hypothesis about why there is a difference in planktonic cell survival between biofilm formation and pre-formed biofilms) in the manuscript text, either in the Results or Discussion sections.

Revision Guidelines

Sincerely,
Anupama Khare
Editor
mSystems

Dear Dr. Anupama Khare and Reviewers:

We really thank you for your carefulness and conscientiousness. Your suggestions are really helpful for revising and improving our paper. According to your suggestions, we have made the following revisions (Changes in the manuscript were marked in green) in this manuscript:

Responds to the reviewers' comments:

Comment: 1) Comment 2 from Reviewer 2 suggested that the panel in Figure 3B-C itself should be labeled. So in the figure itself, for the left column, please add a label 'Inhibition of biofilm formation' and for the right column 'Eradication of pre-formed biofilms'.

Response: Thank you for your suggestions! We have added the labels to Figure 3B-C as required.

Comment: 2) Please include your response to this comment (your hypothesis about why there is a difference in planktonic cell survival between biofilm formation and pre-formed biofilms) in the manuscript text, either in the Results or Discussion sections.

Response: Thank you for your reminder! We have added the following explanation in “Discussion” for differences in resistance of planktonic cells in biofilm eradication and inhibition:

“Significantly, the planktonic cells were more resistant to TAF treatment in biofilm eradication compared to inhibition. The composition of biofilm cells is complex and dynamic (58), the persister cells and metabolically dormant cells exist within the biofilms and are protected by the biofilm matrix (59). We hypothesized that the biofilm matrix was disrupted during biofilm eradication, so that the antibiotic-resistant bacteria were dispatched from the biofilm and became planktonic cells, which showed resistance to TAF.” (linse 486-491).

We sincerely thank the editor and reviewers for their valuable feedback that we have used to improve the quality of our manuscript. Changes in the manuscript were marked in green. We hope the correction will meet with approval. Once again, thank you very much for your comments and suggestions.

Sincerely,

Prof. Wu Yong

E-mail: wuyong_zn@csu.edu.cn

Re: mSystems01026-23R2 (**Repurposing of the anti-malarial agent tafenoquine to combat MRSA**)

Dear Prof. Yong Wu:

Your manuscript has been accepted, and I am forwarding it to the ASM production staff for publication. Your paper will first be checked to make sure all elements meet the technical requirements. ASM staff will contact you if anything needs to be revised before copyediting and production can begin. Otherwise, you will be notified when your proofs are ready to be viewed.

Featured Image Submissions: If you would like to submit a potential Featured Image, please email a file and a short legend to mSystems@asmusa.org. Please note that we can only consider images that (i) the authors created or own and (ii) have not been previously published. By submitting, you agree that the image can be used under the same terms as the published article. File requirements: square dimensions (4" x 4"), 300 dpi resolution, RGB colorspace, TIF file format.

Sincerely,
Anupama Khare
Editor
mSystems